# TOWARDS SPECIFICATION-DIRECTED PROGRAM REPAIR

**Richard Shin**\* **& Dawn Song**
Computer Science Division
University of California, Berkeley
Berkeley, CA 94720, USA
{ricshin,dawnsong}@cs.berkeley.edu

**Illia Polosukhin**
NEAR
illia@near.ai

## ABSTRACT

Several recent papers have developed neural network program synthesizers by using supervised learning over large sets of randomly generated programs and specifications. In this paper, we investigate the feasibility of this approach for *program repair*: given a specification and a *candidate* program assumed similar to a correct program for the specification, synthesize a program which meets the specification. Working in the Karel domain with a dataset of synthetically generated candidates, we develop models that can make effective use of the extra information in candidate programs, achieving 40% error reduction compared to a baseline program synthesis model that only receives the specification and not a candidate program.

## 1 INTRODUCTION

Several recent papers have proposed neural network-based approaches to *program synthesis* from input/output examples (Parisotto et al., 2017; Devlin et al., 2017b; Bunel et al., 2018). In these works, the goal is to generate a program in a domain-specific language from a small number of input/output examples, that can also generalize to unseen examples. The models are trained from a large corpus of programs and a set of input/output examples for each program.

In our work, we consider a variant of the above setting which we dub *specification-directed program repair*, where, in addition to a set of input/output examples that specify the desired semantics, we also get a *candidate* program which we assume is textually similar to a correct program. In real-world applications of program synthesis, we may have access to a slightly erroneous program to use as a candidate, and making effective use of this information would allow us to achieve greater performance than possible with only a specification. For example, many programming tasks involve solving problems similar to existing ones for which we already have solutions.

We demonstrate our methods in the Karel domain, also considered by past work in program synthesis (Devlin et al., 2017a; Bunel et al., 2018). We show that candidate programs indeed contain useful information for synthesizing programs that satisfy I/O examples; our methods achieve 40% error reduction compared to a baseline program synthesis model that does not have access to a candidate program.

Papers such as DeepFix (Gupta et al., 2017), sk_p (Pu et al., 2016), SynFix (Bhatia & Singh, 2016), Wang et al. (2018) and Devlin et al. (2017c) also consider the task of program repair. Unlike these prior works, we focus entirely on semantic errors and do not consider syntactic ones, and consider a very large number of possible task specifications and errors rather than a handful.

## 2 TASK AND MODEL DESCRIPTION

### 2.1 PROBLEM DEFINITION

In the *program synthesis* setting, we are given a set of input/output examples $(I_1, O_1), \cdots, (I_k, O_k)$ and our goal is to find a program $\pi$ such that $\pi(I_1) = O_1, \cdots, \pi(I_k) = O_k$. $k$ is typically small;

---

\*Work partially performed at NEAR.

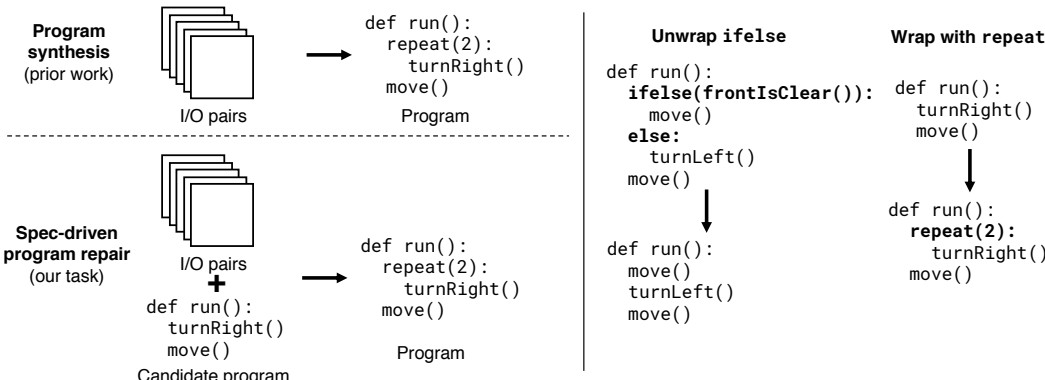

Figure 1: Left: illustration of our task. Right: example mutations used in Section 2.3, to create a synthetic dataset with candidate programs.

for our experiments, we use $k = 5$. In a neural program synthesis approach, we use a dataset containing $N$ programs each with $k$ input/output examples, for training a neural network to represent $p_\theta(\pi \mid (I_1, O_1), \cdots, (I_k, O_k))$ such that it will put high probability on programs which not only satisfy all $(I_i, O_i)$ but will generalize to other I/O examples.

In this work, we consider the setting of *program repair*. In addition to a set of input/output examples $(I_1, O_1), \cdots, (I_k, O_k)$, we are also given $\pi_{cand}$, and our goal is to find a $\pi$ such that $\pi(I_1) = O_1, \cdots, \pi(I_k) = O_k$ and also generalizes to other unseen I/O examples. We assume that there exists such a satisfactory $\pi$ where $\text{dist}(\pi, \pi_{cand}) \leq d$ for some distance measure between programs.

## 2.2 KAREL DOMAIN

Karel is an educational programming language (Pattis, 1981) that has been used in introductory Stanford classes and also in past program synthesis and induction work (Bunel et al., 2018; Devlin et al., 2017a). In Karel, a program consists of instructions for an agent inside a grid. At each cell, the grid (which is between $2 \times 2$ and $16 \times 16$) can contain either an obstacle, or between 0 and 10 markers. The agent starts at some cell in the grid (which may contain markers but no obstacle), and has `move`, `turnLeft`, `turnRight` as actions to move, and `pickMarker`, `putMarker` to manipulate markers. The language contains `if`, `ifElse`, `while` constructs with conditionals {`front`, `left`, `right`}`IsClear`, `markersPresent`, and their negations. `repeat` allows for a fixed number of repetitions. There are no functions or variables.

## 2.3 DATA GENERATION

We used a Karel dataset made publicly available by Devlin et al. (2017a).[1], containing 1,116,854 training examples and 2,500 test examples. Each example is $\{\pi^{(i)}, ((I_1, O_1)^{(i)}, \cdots, (I_6, O_6)^{(i)})\}$; 5 of the I/O pairs are given to the model and the remaining 1 is held out for testing.

In order to create a candidate program $\pi_{cand}^{(i)}$ for training our model, we apply random mutations to $\pi^{(i)}$, where each mutation preserves the syntactic validity of the program. Specifically, we consider the following mutations: insert/delete/replace an action; *wrap* or *unwrap* with `if`/`ifelse`/`while`/`repeat`, sampling a random condition if necessary; replace the conditional/number of repetitions in `if`/`ifelse`/`while`/`repeat`. We consider applying 1, 2, or 3 mutations in sequence to $\pi^{(i)}$, to create $\pi_{cand}^{(i)}$ that differ from $\pi^{(i)}$ by varying amounts.

## 2.4 MODELS

Given a candidate program $\pi_{cand}$ consisting of tokens $t_1, \cdots, t_n$, we first embed each token and then apply a 2-layer bidirectional LSTM to obtain an encoding $e_1, \cdots, e_n$. Following Bunel et al. (2018),

[1] https://msr-redmond.github.io/karel-dataset/

| Model type | Train dist. | $m = 1$, top 1 | | $m = 2$, top 1 | | $m = 3$, top 1 | | $m = 3$, top 64 | |
|---|---|---|---|---|---|---|---|---|---|
| | | Gen. | Exact | Gen. | Exact | Gen. | Exact | Gen. | Exact |
| Tokens | $m = 1$ | 79.28% | 75.88% | 23.92% | 16.56% | 12.52% | 8.12% | 40.28% | 20.72% |
| Tokens | $m = 1, 2$ | 83.24% | **76.16%** | 72.68% | 63.76% | 50.36% | 41.04% | 89.12% | 80.16% |
| Tokens | $m = 1, 2, 3$ | **83.32%** | 73.72% | **77.28%** | **65.92%** | **69.12%** | **57.04%** | **93.96%** | **86.72%** |
| Edits | $m = 1$ | 73.92% | 71.56% | 15.00% | 7.80% | 7.60% | 3.64% | 13.48% | 4.36% |
| Edits | $m = 1, 2$ | 79.40% | 73.36% | 68.24% | 61.00% | 32.32% | 24.28% | 74.32% | 59.40% |
| Edits | $m = 1, 2, 3$ | 79.32% | 71.28% | 70.88% | 61.40% | 60.44% | 50.04% | 92.28% | 84.56% |
| Prog. synthesis baseline | | 70.36 % | 39.04% | 70.36% | 39.04% | 70.36% | 39.04% | 85.80% | 58.12% |
| Random mutations | | 5.08% | 2.56% | 0.72% | 0.24% | 0.4% | 0.04% | 12.24% | 2.92% |

Table 1: Accuracies of our models. $m$ indicates the number of sequential mutations applied to generate candidate $\pi_{\text{cand}}$; columns indicate the value of $m$ at test time. "Gen." means the output program passed all tests and "Exact" means the output program textually matched the $\pi$ in the data. The bottom row shows how often we can recover a correct program when we apply $m$ random mutations to candidate programs; for "$m = 3$, top 64", we attempt $m = 3$ mutations 64 times and report when any of the 64 attempts succeed.

we encode the I/O pairs using a convolutional neural network. An output LSTM which receives the I/O pair embedding $g$ at each timestep creates the synthesized program $\hat{\pi}$.

We considered two variants of the model. In the first (token output), we apply attention over $e_i$ to transfer its information to the output LSTM. At each step, we compute $h_i, \tilde{o}_i = LSTM(h_{i-1}, [t_{i-1} \ s_{i-1} \ g]); s_i = Attention(\{e_1, \cdots, e_n\}, \tilde{o}_i); o_i = W_o[\hat{o}_i \ s_i]^T + b; p_i = Softmax(W_p o_i)$. $h_0$ (initial LSTM state) and $s_0$ (initial attention context) are set to 0. $p_i \in [0, 1]^{|V|}$ is a probability distribution over over possible program tokens.

In the second (edit output), we explicitly make use of the fact that we would like to make small modifications to existing code, and train the model to output *edits*. At step $i$, we have a pointer $k_i$ to the tokens of $\pi_{\text{cand}}$, and choose an action among *keep* (copy $\pi_{\text{cand}}[k_i]$ to the output), *delete*, *insert* a token $t$, or *replace* $\pi_{\text{cand}}[k_i]$ with $t$; there are $2 + 2|V|$ actions where $|V|$ is the vocabulary size. We set $k_{i+1} = k_i$ unless the last action was *insert*. The computation each step is $h_i, o_i = LSTM(h_{i-1}, [t_{i-1} \ a_{i-1} \ e_{k_i} \ g]); t_{i-1}$ is an embedding for the program token from the last step (or a null token if the last action was *delete*) and $a_{i-1}$ is the last edit operation (one of $2 + 2|V|$). Because there is no need to compute attention weights in this approach, the forward pass takes $O(q + p)$ rather than $O(q + q \cdot p)$ time, where $q$ and $p$ are the number of tokens in the candidate and true programs, respectively. In practice, we found it is 3–4× faster to train than the first variant.

## 3 EXPERIMENTAL RESULTS

Table 1 summarizes our main experimental results. To measure the value of using $\pi_{\text{cand}}$, we compare against our baseline program synthesis model (which does not have access to $\pi_{\text{cand}}$). When using $\pi_{\text{cand}}$ with $m = 1$, our best program repair model achieves 83.32% top-1 generalization accuracy compared to 70.36% of the program synthesis model; a 40% reduction in error rate. When $m = 3$ (and therefore $\pi_{\text{cand}}$ is less useful), the top-1 generalization accuracy of 69.12% is lower than the baseline's 70.36%, but the exact match rate of 57.04% is higher than the baseline's 39.04%; top-64 generalization accuracy of 93.96% also beats the baseline's 85.80%.

The edit-output model performs similar to but consistently worse than the token-output model, although it has the advantage of lower computational cost. Interestingly, when testing with $m = 1$ mutations to create $\pi_{cand}$, the models trained with the exact same data distribution ($m = 1$) perform worse than those trained with a wider variety of mutations ($m = 1, 2, 3$). We also found that using the $e_i$ (embeddings of the candidate program) at each step of decoding was highly critical for the model to work; variants without (not in table) either failed to learn or achieved very similar results to the baseline program synthesis model, indicating that it is not using information from the candidate program.

For all methods, we used a beam search with beam size 64. For top 1 accuracy, we only consider the most likely output of the beam search; for top 64, we check whether any of the 64 outputs are correct.

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
