# OpenReview forum: "Towards Specification-Directed Program Repair"
_ICLR.cc/2018/Workshop — Accept_

### Official Review · AnonReviewer3 · 2018-03-09
**Semantic program repair is not novel**

**Rating:** 7
**Confidence:** 4

**Review:**

This paper proposes using neural networks for program repair. In particular, given

1) a grammar for programs
2) some input/output pairs for correct program behavior
3) a faulty candidate program that almost takes inputs into outputs

Recover the correct program through some small changes. In particular, the paper restricts itself to faulty programs which are syntactically valid but have made a semantic error. This is an interesting and fascinating area with lots of interesting work, much of it referenced by this paper.

Unfortunately, this is not a particularly novel problem as it has been explored by Gulwani et al. (https://arxiv.org/abs/1603.03165) as well as Wang et al.

This paper would be much stronger if it could be compared something like Wang et al. at least on the same set of problems where the mutations were restricted to semantic changes.

The paper could be written a bit more clearly, but was fairly easy to follow.

---

### Official Review · AnonReviewer1 · 2018-03-10
**Program repair using synthesis and edit-model**

**Rating:** 7
**Confidence:** 4

**Review:**

This paper presents two neural architectures for modeling the program repair problem, where the problem is to modify a given candidate program (close to a correct desired program) such that it becomes consistent with a given set of input-output examples. Previous neural program approaches either perform syntactic repairs or use constraint-based techniques for semantic repairs. It was great to see an end-to-end neural approach also aiming to tackle semantic program repair with test-cases.

The problem formulation of using an end-to-end neural approach to repair programs with test-cases (i/o examples) is novel. This approach also has the potential to improve program synthesis techniques. The results presented generally show the benefits of access to candidate programs (although the case with m=3 is interesting where it performs slightly worse than the baseline).

Given the shorter page limit, it might be difficult to present all the information. But it might be good to provide a bit more information about how different types of mutations effect the final results (e.g. does modifying control flow statements result in different performance than modifying action statements?). Also, it wasn't clear if the mutations involved dropping control flow statements such as if, while etc. It would be nice to quantify the space of possible mutations to make it clear than simple search methods for mutations won't scale to this problem.

Are there some intuitions why the edit-model performed worse than the LSTM generative model? It seems the edit-model needs to perform lesser (easier) work. It might also be interesting to investigate why training on m=1,2,3 results in better performance on m=1 test set compared to training on m=1 training set.

Minor corrections

page 3: distribution over over possible --> distribution over possible
page 3: computation each step is --> computation at each step is

---

### Official Review · AnonReviewer2 · 2018-03-12
**.**

**Rating:** 6
**Confidence:** 4

**Review:**

This paper solves program repair which is defined as follows: given input/output pairs of a true program pi, and a candidate program pi_cand, get back pi. A program is represented as a string of tokens. The Karel dataset (with the corresponding language) is used. Training data is generated synthetically.

A neural net is trained to map from IO pairs and pi_cond to 1) pi or 2) edits to pi_cond (to obtain pi). This method is better than the baseline (where pi_cond) is not used.

The contribution seems a bit marginal because the domain language is small and it is not very surprising that this method beats the baseline. However, it is nice as a workshop paper.

---

### Decision · Program_Chairs · 2018-03-20
**ICLR 2018 Workshop Acceptance Decision**

**Decision:**

Accept

**Comment:**

Congratulations, your paper was accepted to the ICLR workshop.